# Targeted Epigenetic Interventions in Cancer with an Emphasis on Pediatric Malignancies

**DOI:** 10.3390/biom13010061

**Published:** 2022-12-28

**Authors:** Zsuzsanna Gaál

**Affiliations:** Department of Pediatric Hematology-Oncology, Institute of Pediatrics, University of Debrecen, 4032 Debrecen, Hungary; zsuzsanna.gaal.46@gmail.com

**Keywords:** pediatric cancer, epigenetics, hypomethylating agents, histone modifiers, nucleosome remodeling, clinical translation

## Abstract

Over the past two decades, novel hallmarks of cancer have been described, including the altered epigenetic landscape of malignant diseases. In addition to the methylation and hyd-roxymethylation of DNA, numerous novel forms of histone modifications and nucleosome remodeling have been discovered, giving rise to a wide variety of targeted therapeutic interventions. DNA hypomethylating drugs, histone deacetylase inhibitors and agents targeting histone methylation machinery are of distinguished clinical significance. The major focus of this review is placed on targeted epigenetic interventions in the most common pediatric malignancies, including acute leukemias, brain and kidney tumors, neuroblastoma and soft tissue sarcomas. Upcoming novel challenges include specificity and potential undesirable side effects. Different epigenetic patterns of pediatric and adult cancers should be noted. Biological significance of epigenetic alterations highly depends on the tissue microenvironment and widespread interactions. An individualized treatment approach requires detailed genetic, epigenetic and metabolomic evaluation of cancer. Advances in molecular technologies and clinical translation may contribute to the development of novel pediatric anticancer treatment strategies, aiming for improved survival and better patient quality of life.

## 1. Introduction

The five year overall survival (OS) of children diagnosed with cancer increased from 58% to 84.7% between the mid-1970s and mid-2010s, respectively [1,2]. The cancer mortality rate among pediatric and adolescent patients has declined by more than 50% over the last 50 years [2]. However, chemoresistance and toxicity are still responsible for many deaths. Epigenetic alterations are heritable, and reversible changes in gene expression pattern [3] offer promising opportunities for clinical translation (Figure 1).

Mutations and translocations of genes encoding epigenetic enzymes and histone proteins are frequently identified in cancer, such as TET2 and DNMT3A mutations, translocations of MLL and NUP98 genes in acute myeloid leukemia (AML), mutations of ARID1A and SMARCA4 genes in gliomas, SETD2 mutations in clear cell renal cell carcinoma (ccRCC), and oncohistone mutations in pediatric brain and bone tumors [4,5,6,7,8,9,10,11,12,13,14,15]. Malignant diseases also feature alterations in their epigenetic profiles and altered expression of epigenetic modifiers [16,17,18,19], which can be applied as biomarkers in differential diagnosis, chemoresistance prediction and risk stratification [20] (Table 1). 

According to DNA-methylation profiles, T-cell lymphoblastic lymphoma and pilocytic spinal cord astrocytoma can be distinguished from T-ALL and diffuse leptomeningeal glioneuronal tumors, respectively [21,22]. The methylation level of TFAP2A in circulating tumor DNA is a diagnostic biomarker for retinoblastoma [23]. Differentially methylated positions are promising markers for platinum chemotherapy resistance in various cancers [24]. The expression level of SIRT2 has a positive relationship with cytosine arabinoside and daunorubicin resistance in AML cells [25]. The activation of the EZH2-Stat3 signaling axis is implicated in the development of temozolomide resistance in glioblastoma [26]. The loss of TET2 and reduced levels of genomic 5-hydroxymethylcytosine (5hmC) are associated with poor survival in AML [27]. NUP98/NSD1 fusion is strongly associated with adverse prognosis in pediatric AML [28]. In juvenile myelomonocytic leukemia (JMML), a high level of DNA methylation indicates high relapse incidence and inferior OS, while a low methylation level is associated with a favorable outcome [29]. EZH2 mutation and SETD2 deficiency correlates with poor survival in myelodysplastic syndrome (MDS) [30,31]. A low level of SIRT6 predicts poor relapse-free survival (RFS) in Hodgkin lymphoma [32]. A high level of methylated O^6^-methylguanine-DNA methyltransferase promoter methylation is associated with a favorable outcome of medulloblastoma [33], loss of 5hmC correlates with adverse prognosis in WHO grade II diffuse astrocytomas [34], while EZH2 expression has been described as an independent marker of poor prognosis in pediatric ependymoma [35]. 5hmC profiles correlate with metastatic burden in neuroblastoma [36]. Higher levels of DOT1L expression are associated with poor OS and RFS in ccRCC [37]. The DNA methylation profile has been described as an independent prognostic biomarker for pediatric adrenocortical tumors [38].

In this review, possibilities of targeted epigenetic interventions are discussed in the most common pediatric malignancies, including acute leukemias, lymphomas, neuroblastoma, gliomas, soft tissue sarcomas and kidney tumors (Table 2). The vast majority of cited publications and clinical trials summarize experiences with pediatric and adolescent patients. However, since clinical experience can be first obtained in adult cancer patients (following successful in vitro and in vivo experiments), some data originating from cell lines, animal models and adult patients are also included in order to highlight the wide variety of novel opportunities. Major groups of epidrugs are discussed based on the epigenetic modifications that they target. 

## 2. DNA Methylation

Methylation of the fifth carbon of cytosines is catalyzed by DNA methyltransferase (DNMT) enzymes, resulting in the formation of 5-methylcytosine (5mC) and transcriptional repression [39]. DNMT1 and DNMT3 are canonical DNMTs that catalyze maintenance and de novo DNA methylation, respectively [40]. Azacitidine and 5-aza-2′-deoxycytidine (decitabine) are cytosine analogs, commonly referred to as hypomethylating agents (HMA) that inhibit DNMT enzymes [41].

### 2.1. HMA and Hematological Malignancies

Azacitidine and decitabine have been approved for therapy of MDS and AML since 2004 and 2006, respectively [42]. According to single-center pediatric experience with venetoclax+azacitidine treatment, morphologic response and MRD negativity have been achieved in 2/2 high-grade MDS and 4/6 AML patients, respectively [43]. Compared to AML-type chemotherapy, a much better survival rate was registered with a decitabine-combined minimally myelosuppressive regimen bridged with allo-HSCT in children with advanced MDS [44]. Successful administration of HMAs have been reported in the case of three adolescent AML patients with monosomy 5/del(5q), who received decitabine treatment during both remission induction and conditioning, and were free of disease at 3.6, 3.2, and 3.0 years after HSCT, respectively [45]. According to a phase 1 clinical trial (NCT01861002) published in 2018, azacitidine can be used safely in sequence with intensive chemotherapy in relapsed/refractory pediatric AML and offers encouraging clinical activity [46]. An infant with JMML, harboring somatic KRAS mutation and monosomy 7, was reported in 2019 who achieved sustained remission following azacitidine monotherapy [47]. An adolescent patient with chronic myelomonocytic leukemia (CMML) received venetoclax/decitabine treatment as a bridge to HSCT [48]. Decitabine has shown antineoplastic activity in ALK+ anaplastic large cell lymphoma (ALCL) cell lines [49]. Moreover, according to in vitro and in vivo results, low-dose crizotinib with decitabine treatment completely suppressed the emergence of resistant cells in ALK+ ALCL [50].

### 2.2. HMA and Solid Tumors 

According to recently published data, DNMT3A inhibitor SGI-1027 repressed the development of glioblastoma by indirect inhibition of the TGF-β signaling pathway [51]. In relapsed or progressive rhabdoid tumors, radiological signs of antitumor activity have been registered following decitabine-augmented chemotherapy [52]. Decitabine was found to potentiate the cytotoxic effects of cisplatin in neuroblastoma cells by the induction of RIG-I-related innate immune response [53]. HMA treatment with decitabine increased the susceptibility of rhabdomyosarcoma, Ewing sarcoma and osteosarcoma cell lines to cytotoxic T-lymphocyte mediated lysis [54]. Furthermore, treatment of 143B osteosarcoma cells with decitabine resulted in the inhibition of osteosarcoma growth and metastasis by enhanced expression of ERα [55]. In synovial sarcoma cell lines, decitabine is also suggested to have good therapeutic potential [56]. The Ras association domain-containing protein 1 isoform A (RASSF1A) promoter hypermethylation might be involved in the development and aggressiveness of some pediatric renal tumors and correlated with a poor prognosis. Hypermethylation in the RASSF1A promoter region in rhabdoid tumor of the kidney and ccRCC is associated with aggressive disease and poor prognosis, which could successfully be reversed by the administration of decitabine treatment [57].

## 3. Targeting DNA-Hydroxymethylation

TET proteins are methylcytosine dioxygenase enzymes that catalyze the oxidation of 5mC to 5-hydroxymethylcytosine (5hmC) [58]. TET2 is a master epigenetic regulator of hematopoiesis whose function is inhibited by the recurrent mutations of isocitrate dehydrogenase (IDH) enzymes, resulting in the formation of the oncometabolite 2-hydroxyglutarate [59]. Therefore, enhancing TET2 enzymatic activity or restoring *TET2* transcription may be clinically beneficial in hematological malignancies [5]. On the other hand, the TET1/2 inhibitor (Bobcat339) has been shown to reduce T-ALL burden by targeting the dependence of T-lymphoblasts on the tricarboxylic acid cycle for their growth and survival [60,61]. In glioma cell lines, overexpression of TET3 partially restored the genome-wide 5hmC patterns of control brain samples [62]. Pharmacological inhibition of TET1 reduced cell viability in a mouse model of medulloblastoma [63]. In osteosarcoma cells, enhanced expression of TET1 was associated with an increase in apoptosis rate [58], while the inhibition of TET2-dependent induction of IL-6 is a potential therapeutic approach through antagonizing metastasis formation [64].

## 4. Histone Code

Nucleosomes are composed of an octamer of four core histones (H3, H4, H2A and H2B), wrapped with 147 base pairs of DNA [65]. N-terminal histone tails are enriched with a variety of posttranslational modifications (PTMs) [66]. Histone modifications may result in both activation and repression of transcription, controlled by an array of histone modifiers. The balance between activating and repressing histone marks is commonly disrupted in malignant diseases that can be therapeutically targeted.

### 4.1. Histone Acetylation

Histone acetyltransferase (HAT) enzymes acetylate ε-amino groups of histone lysine residues by three major HAT families, namely p300/CREB-binding protein (p300/CBP), MYST (Moz, Ybf2, Sas2 and Tip60) and GNAT (GCN5-related N-acetyltransferase) [67], described as critical regulators of cell development and carcinogenesis [68]. Histone lysine acetylation results in the activation of transcription.

GCN5 inhibitor α-methylene-γ-butyrolactone 3 decreased acetylation and protein level of the chimeric transcription factor E2A-PBX1 in pediatric pre-B-cell ALL with t(1;19) translocation [69]. In non-APL AML, GCN5 was described as a potential therapeutic target through its contribution to ATRA resistance via aberrant acetylation of histone 3 lysine 9 (H3K9ac) residues [70]. Inactivation of MOF enzyme (MYST1) suppressed leukemia development in a NUP98-HOXA9-driven AML model [71]. Activators of p300 can be applied in the treatment of MDS with TET2 inactivating mutations in order to suppress its transition to AML [72]. In Burkitt lymphoma, inhibition of GCN5 attenuated BCR signaling and reduced the tumorigenic properties of cells [73]. Epigallocatechin-3-gallate blocks p300-mediated acetylation of p65 protein, thereby impairing the transformation of B-cells by EBV [68]. 

Caspase-independent cell death was triggered in neuroblastoma cell culture by PU139 (pan-inhibitor of HAT enzymes) [74]. Silencing or pharmacological inhibition of PCAF in alveolar rhabdomyosarcoma resulted in the down-regulation of PAX3-FOXO1 reduced proliferation and tumor burden in xenograft mice models [75]. Inhibition of the HBO enzyme (MYST2) via intraperitoneal injection of a single dose of WM-3835 potently suppressed the growth of an osteosarcoma xenograft in SCID mice [76]. Tip60 is a novel therapeutic target in osteosarcoma by promoting the effects of KDM2 acetylation on proliferation and metastasis formation of tumor cells [77].

### 4.2. Histone Deacetylation

According to their sequence similarities with yeast enzymes, 18 human zinc-dependent histone deacetylases (HDACs) have been identified [78,79]. The first clinically successful HDAC inhibitor, suberoylanilide hydroxamic acid (SAHA/vorinostat), was approved by the FDA in 2006 as a treatment for refractory or relapsed cutaneous T-cell lymphoma [80]. 

#### 4.2.1. HDACi and Hematological Malignancies

According to a phase 1 clinical trial that was performed with panobinostat in 2020 (NCT02676323), 8/17 pediatric patients with relapsed or refractory (R/R) AML achieved complete remission (CR), and no dose-limiting toxicities were observed [81]. In a phase I trial (NCT02412475) published in 2022, decitabine and vorinostat was well-tolerated and effective in R/R pediatric AML patients in combination with fludarabine, cytarabine and G-CSF (FLAG) treatment [82]. In the MLL-AF9 driven MOLM-13 cell line, mocetinostat reduced cell viability and induced apoptosis [83], while HDAC inhibitor I13 could be a potent and selective agent in AML patients with t(8;21) translocation or MLL-rearrangement to surmount differentiation block [84]. Belinostat induced granulocytic differentiation of acute promyelocytic leukemia HL-60 cells more effectively compared with retinoic acid treatment alone, that was associated with histone H4 hyperacetylation of the C/EBPα promoter region [85]. In t(4;11)-positive primary infant ALL cells, HDAC inhibitors including trichostatin A, vorinostat, panobinostat, valproic acid and romidepsin effectively induced leukemic cell death accompanied by the downregulation of MYC proto-oncogene as well as the MLL-AF4 fusion product [86]. In a large phase II clinical study (NCT00742027) in patients with relapsed classical Hodgkin’s lymphoma, panobinostat reduced tumor measurements in 74% of patients, including 23% partial remissions and 4% CR [87]. Romidepsin has been shown to potentiate the anti-tumor effect of anti-CD20 chimeric antigen receptor (CAR) modified expanded peripheral blood natural killer (NK) cells against rituximab-sensitive and -resistant Burkitt lymphoma in immunodeficient mice [88]. Cay10603, a potent HDAC6 inhibitor inhibited cell cycle progression in Burkitt lymphoma cell lines [89].

#### 4.2.2. HDACi and Solid Tumors

HDACi SAHA, sodium butyrate and trichostatin A induced apoptosis related to dissipation of mitochondrial membrane potential and activation of caspase enzymes in Daoy and UW228-2 medulloblastoma cells [90]. HDAC6-selective inhibitors demonstrated in vitro therapeutic potential against group 3 medulloblastoma [91]. Antiproliferative effects of vorinostat and mocetinostat were described in different human glioma cells [92]. Unfortunately, vorinostat failed to improve outcome in childhood diffuse intrinsic pontine glioma (DIPG) [93]. On the other hand, treatment of group C ependymoma DKFZ-EP1NS cells with vorinostat induced neuronal differentiation and loss of stem cell-specific properties [94]. OBP-801 (spiruchostatin A) induced G2/M phase arrest and suppressed tumor growth in human neuroblastoma cells and in a mouse xenograft model, respectively [95]. Selective dose-dependent cytotoxicity of romidepsin (depsipeptide) was described in both single copy and N-myc amplified neuroblastoma cell lines [96]. Panobinostat enhanced NK cell cytotoxicity in soft tissue sarcoma cell lines [97], while targeting HDAC6 is a promising therapeutic option in rhabdomyosarcoma [98]. Selective inhibition of HDAC6 induced the downregulation of EWSR1-FLI1 and significantly reduced its oncogenic functions in Ewing sarcoma cell lines [99]. Pan-HDAC inhibitor LBH589 (panobinostat) treatment resulted in apoptosis induction and inhibition of cell proliferation of SK-NEP-1 and G401 Wilms’ tumor cells. LBH589 also had a significant effect in SK-NEP-1 xenograft tumors [100]. Targeting HDAC1 is a promising therapeutic approach in aggressive hepatoblastoma due to its involvement in the repression of p21 [101]. In retinoblastoma, intravitreal belinostat was equally effective as standard-of-care melphalan but without retinal toxicity [102]. CUDC907, a dual phosphoinositide-3 kinase/HDAC inhibitor, promoted apoptosis of NF2 schwannoma cells [103].

### 4.3. Sirtuin Enzymes

Sirtuin (SIRT) enzymes compose an NAD^+^-dependent class III subfamily of HDAC enzymes with characteristic intracellular localizations and variable enzyme activities that are also key players in cancer cell metabolism [104]. Seven members of the family have been identified (SIRT1-7), among which SIRT1 was the first sirtuin to be shown to be involved in cancer [105].

Although tenovin-6-mediated SIRT1/2 inhibition hampered the growth of primary cells from children with ALL [106], and silencing of SIRT1 prolonged the lifespan in a mouse model of T-ALL [107], the growth of leukemic cells was promoted by SIRT1 inhibition and reduced by SIRT1 activation in two T-ALL cell lines carrying Notch mutations [108]. Inhibition of SIRT1 reduced proliferation of AML cell lines with t(8;21) translocation [109]. NRD167, an inhibitor of SIRT5 enzyme, reduced glutamine utilization and induced apoptosis in primary AML samples and cell lines [110]. Downregulation of the SIRT1 protein has been described in MDS stem and progenitor cells, whereas activation of SIRT1 represents a promising means to target MDS [111]. Inhibition of SIRT1 deprived Burkitt lymphoma cells of their most important survival signal by reducing MYC protein levels [112].

In patient-derived IDH mutant glioma lines, overexpression of SIRT1 led to inhibition of cell growth [113]. On the other hand, specific inhibition of SIRT1 by EX527 induced cell apoptosis in two glioma cell lines by activating p53 [114]. Based on its interaction with the ERK/STAT3 signaling pathway, SIRT7 may also function as a valuable target for the treatment of human glioma [115]. Downregulation of SIRT3 was associated with suppression of growth and migration of glioblastoma cells [116]. Overexpression of SIRT4 significantly reduced proliferation [117], while inhibition of SIRT6 induced differentiation of neuroblastoma cell lines [118]. Pharmacological inhibition of SIRT1 and SIRT2 impaired autophagy process and induced cell death in pediatric soft tissue sarcoma cell lines [119]. SIRT1i molecules are promising therapeutic targets to treat metastatic disease in Ewing sarcoma by mediating tumor suppressive Notch response [120]. In ccRCC cells, overexpression of SIRT4 reduced proliferation and migration [121], while inhibition of SIRT6 may counteract the Bcl-2-dependent pro-survival pathway [122]. Overexpression of SIRT6 induced apoptosis of nasopharyngeal carcinoma cells by inhibiting NFκB signaling [123].

### 4.4. Histone Methylation

Histone methylation occurs on arginine and lysine residues, resulting in different outcomes for transcriptional regulation [66]. Lysines can be mono-, di- or trimethylated on their ε amine group, arginines can be monomethylated (MMA), symmetrically dimethylated (SDMA) or asymmetrically dimethylated (ADMA) on their guanidinyl group, the processes of which are believed to turnover more slowly than many other PTMs [124]. Three families of “writer” histone methyltransferases are distinguished based on their domain structure and the methylated residue [125]. Vast majority of enzymes that catalyze histone lysine methylation (KMT) contain the so-called SET domain, except for the DOT1L enzyme. The third group is composed of histone arginine methyltransferase PRMT enzymes. Based on the amount of recently published results, EZH2 is discussed separately from other SET-domain containing KMT enzymes.

#### 4.4.1. EZH2i

EZH2i 3-deazaneplanocin A induced growth inhibition and increased apoptotic rate in T-ALL Jurkat cells [126]. EZH1/2 dual inhibitor, DS-3201 inhibited the growth of B-ALL, harboring MLL-AF4 significantly in a patient-derived xenograft mouse model [127]. Proliferation of pediatric acute monocytic leukemia cells was inhibited by targeting EZH2-mediated methylation of histone H3 [128]. Targeting EZH2 in myelodysplasia is a promising treatment strategy, since it promoted the transformation from MDS to AML [129]. Tazemetostat (EPZ-6438) lead to potent antitumor activity in preclinical models of EZH2-mutant non-Hodgkin lymphoma (NHL) [130]. In a first-in-human, open-label, phase 1 study (NCT01897571), published in 2018, tazemetostat showed a favorable antitumor activity in refractory B-cell NHL and advanced solid tumors [131]. Valemetostat is the first dual EZH1/2 inhibitor, approved for the treatment of adult T-cell lymphoma in September 2022 [117].

Pharmacological inhibition of EZH2 impaired proliferation and induced apoptosis of SHH medulloblastoma cells in vitro [132]. Based on its interaction with the tumor suppressor protein p16, EZH2 is a potential therapeutic target for H3K27M-mutant pediatric gliomas [133]. EZH2i GSK343 led to significantly decreased viability, migration and invasion in neuroblastoma cell lines [134], while EPZ005687 reduced cell viability and colony formation in rhabdomyosarcoma cell lines [135]. Based on the positive results of a single-arm phase II basket study (NCT02601950), the FDA granted accelerated approval of tazemetostat in 2020 for patients aged 16 years and older with metastatic or advanced epithelial sarcoma not eligible for complete resection [136]. EZH1/2 dual inhibitors are promising therapeutic strategies for pediatric malignant rhabdoid tumors [137].

#### 4.4.2. Other SET-Domain-Containing KMT Enzymes

Covalent inhibition of the NSD1 enzyme impaired colony formation of primary AML cells harboring t(5;11) translocation and NUP98-NSD1 fusion that is predominantly observed in pediatric AML patients [28,138]. In childhood ALL cells, inhibition of G9a (EHMT2) with BIX01294 abrogated transendothelial migration [139]. In cultured cells of AML and ALL, inhibition of G9a reduced cell proliferation and promoted apoptosis [140]. Inhibition of SUV39H1 reduced proliferation and cell migration in pediatric astrocytoma cell lines [141]. In pediatric-type high-grade glioma cells, inhibition of SUV39H1 and SUV39H2 methyltransferases was confirmed to be lethal [142]. Treatment of neuroblastoma cells with BIX01294 resulted in the inhibition of cell growth and proliferation [143]. G9a was also described as a potential therapeutic target in embryonal rhabdomyosarcoma due to its interaction with Wnt signaling [144]. Inhibition of G9a reduced metastatic development in mice models of Ewing sarcoma [145]. Knockdown or inhibition of SUV39H1 suppressed the growth of ccRCC cells by inducing ferroptosis [146].

#### 4.4.3. DOT1L

DOT1L is the only KMT enzyme that does not contain a SET domain and specifically targets histone H3 lysine 79 (H3K79) residues for mono-, di- or trimethylation [147]. MLL-rearranged acute leukemias are confirmed to be dependent on aberrant H3K79 methylation by the DOT1L enzyme [148]. Inhibition of DOT1L with pinometostat (EPZ5676) resulted in significant differentiation effects in MLL-fused leukemia cell lines [149]. Based on the results of a phase 1 clinical trial in pediatric R/R leukemia patients with MLL-rearrangement, performed between 2014 and 2016 (NCT02141828), administration of EPZ5676 in combination with other antileukemia agents is warranted [150]. In vivo efficacy of DOT1L inhibition has also been observed in a nude rat xenograft model of *DNMT3A*-mutant AML [151]. The inhibition of DOT1L with SGC0946 reduced H3K79 methylation and proliferation of N-Myc amplified neuroblastoma cells [152]. In murine orthotopic xenografts of retinoblastoma, EPZ5676 significantly improved treatment efficacy [153].

#### 4.4.4. PRMT Enzymes

Among the nine PRMTs described, type I, II and III enzymes are able to generate ADMA, SDMA or MMA, respectively [154]. Maintenance of FLT3-ITD AML cells was markedly blocked by genetic or pharmacological inhibition of PRMT1 enzyme [155]. Moreover, inhibition of PRMT1 with MS023 abolished arginine methylation of FLT3 and disrupted the maintenance of MLL-rearranged ALL cells [156]. Spliceosomal mutant leukemias were found to be preferentially sensitive to PRMT inhibition [157]. 28d, a potent inhibitor of type I PRMTs, effectively inhibited cell proliferation in several types of leukemia cell lines [158]. Inhibition of PRMT5 induced cell death in different types of NHL cell lines through the abrogation of proliferation-related signaling pathways [159]. The depletion of PRMT1 induced apoptosis in medulloblastoma cells [160], while treatment with PRMT5 inhibitor decreased tumor growth and increased survival in a SHH medulloblastoma mouse model [161]. AMI-1 and SAH, pan-inhibitors of PRMT enzymes, decreased cell viability and reduced the invasive phenotype of rhabdomyosarcoma cells [162].

### 4.5. Histone Demethylation

Histone lysine demethylases are categorized into two subgroups, KDM1-family (KDM1A = LSD1 and KDM1B) and Jumonji C (JmjC) domain-containing histone demethylase enzymes (JHDMs) [163]. Arginine demethylation occurs via peptidyl arginine deiminase 4 by converting arginine to citrulline [164].

#### 4.5.1. KDM1-Family 

The knockdown or chemical inhibition of LSD1 dominated C/EBPα instead of the GATA1 transcription factor, resulting in metabolic shifts and growth arrest in erythroleukemia cells [165]. Inhibition of LSD1 by the highly potent FY56 compound induced differentiation in MOLM-13 and MV4-11 AML cell lines [166]. LSD1 inhibitor S2157 has been confirmed to efficiently pass through the blood–brain barrier and eradicate CNS leukemia in T-ALL mice models [167]. According to xenograft studies with patient-derived Ewing sarcoma cell lines, LSD1 inhibitor HCI2509 disrupted the oncogenic transcriptional activity of EWS/ETS fusion proteins [168]. In retinoblastoma cells, inhibition of LSD1 by SP2509 resulted in growth inhibition via the suppression of β-catenin pathway [169].

#### 4.5.2. JHDM Enzymes

According to in vitro and in vivo biological function experiments, KDM4A (JMJD2A) inhibitor SD49-7 suppressed the progression of leukemia stem cells through the activation of the apoptosis signaling pathway [170]. GSKJ4, an inhibitor of KDM6A (UTX) and KDM6B (JMJD3) enzymes, induced apoptosis and cell-cycle arrest in Kasumi-1 cells, decreased proliferation of U-937 and K-562 cells, and attenuated disease progression in a human AML xenograft mouse model [171,172]. Inhibition of KDM3C (JMJD1C) by JDI-10 decreased lipid synthesis-associated genes and induced apoptosis in MLL-rearranged AML cells [173]. Inhibition of KDM5A (JARID1A) greatly potentiated the differentiation of APL cell line NB4 [174]. Genetic and pharmacologic inhibition of KDM4B (JMJD2B) substantially delayed tumor growth in preclinical subcutaneous xenograft models of PAX3-FOXO1-driven alveolar rhabdomyosarcoma [175]. Depletion of KDM3A (JHDM2A) also inhibited growth and metastasis formation of the oncofusion-positive rhabdomyosarcoma cells in vivo [176]. KDM5B (JARID1B) inhibitor AS-8351 suppressed proliferation and induced cell cycle arrest in Ewing sarcoma cell lines [177]. Knockdown of KDM3A suppressed aerobic glycolysis and weakened the growth of osteosarcoma cells in vitro and in a nude mouse model [178]. Downregulation of JMJD6 enzyme resulted in impaired colony formation of ccRCC cells [179]. SMARCA4-deficient tumors are confirmed to be strongly dependent on KDM6A and KDM6B histone demethylases, which are also novel promising therapeutic targets [180].

## 5. Reader Molecules

Numerous chromatin-associated factors can specifically interact with methylated CpG dinucleotides and modified histones via distinct domains, which are essential for the assembly of multiprotein epigenetic regulator complexes. Over the previous decade, growing numbers of reports were published about successful inhibition of so-called reader proteins in cancer (Figure 2).

### 5.1. DNA Methylation Readers

Methyl-CpG binding zinc finger proteins, MBD-containing proteins and SRA domain-containing proteins are the three subgroups of methyl-binding proteins (MBPs), among which methyl-CpG-binding protein 2 (MeCP2) was the first MBD-containing protein discovered in 1992 [181,182]. Deletion of MBD2 effectively switched off the abnormal activation of Wnt signaling in T-ALL cell lines and mice models [183]. Knockdown of UHRF1 protein reduced c-Myc protein expression and cell viability in both B-ALL and T-ALL in vitro [184]. Overexpression of MeCP2 in C6 glioma cells resulted in decreased proliferation, migration and invasion [185]. In primary glioblastoma tumor samples, knockdown of MBD2 restored expression of the tumor suppressor BAI1 protein [186]. According to functional experiments, overexpression of CTCF could inhibit migration and invasion of ccRCC cells [187].

### 5.2. Acetyl-Lysine Readers

Reader proteins of lysine acetylation contain bromodomain (BRD), double plant homeodomain (PHD) fingers or a YEATS (*Yaf9*, ENL, AF9, Taf14, Sas5) domain, among which bromodomain and extra terminal (BET) proteins compose a distinct subfamily of BRD group [188,189,190]. Inhibition of BRD4 by I-BET151 efficiently blocked proliferation of AML cells in primary murine hematopoietic stem and progenitor cells harboring t(10;17)(p15;q21) translocation [191]. TDI-11055, an orally bioavailable small-molecule inhibitor of ENL, blocked disease progression in patient-derived xenograft models of NPM1-mutated and MLL-rearranged AML [192]. Inhibition of MOZ (MYST3) arrested tumor growth and induced senescence in mice models of lymphoma [193]. Bromodomain inhibitor OTX015 led to downregulation of MYC and cell cycle arrest in ALK+ ALCL cells [194]. BET inhibitor JQ1 was found to potently decrease viability of MYC-amplified medulloblastoma cells [195]. In mouse glioblastoma cells, OTX015 showed much higher antiproliferative effect compared to that of JQ1 [196]. Targeted inhibition of BRD4 reduced cell proliferation and invasiveness of ATRT cell lines [197]. OTX015 treatment resulted in reduced proliferation and upregulation of apoptosis-related proteins in pediatric patient-derived ependymoma stem cell models [198].

### 5.3. Methyl-Lysine Reader Domains

Major families of methyl-lysine reader domains include chromodomains (CD), PHD, WD40 repeat (WDR) and PWWP (proline–tryptophan–tryptophan–proline) domains [199]. CD inhibitor SW2_110A inhibited proliferation of THP1 leukemia cells [200]. Deletion of WDR5 impaired colony forming ability of MLL-AF9 positive cells in a murine leukemia model [201]. PHF6 and NSD2 have been described as promising therapeutic targets in PHF6-mutant AML and ALL, respectively [202,203]. Overexpression of CBX7 inhibited cell proliferation, migration and colony formation of glioma cell lines [204].

## 6. Targeting Nucleosome-Remodeling Machinery

Four major subfamilies of chromatin remodeling complexes have been identified, switch/sucrose non-fermentable (SWI/SNF), imitation SWI (ISWI), chromodomain-helicase DNA-binding protein (CHD) and inositol-requiring mutant 80 (INO80), which are responsible for mobilization of nucleosomes at target-promoters and -enhancers to modulate gene expression (Figure 3) [10,205,206,207].

Dual inhibitors of SWI/SNF catalytic subunits, BRM (SMARCA2) and BRG1 (SMARCA4) lead to downregulation of leukemic pathway genes, including MYC in AML cell lines [208]. Deletion of SMARCA5, catalytic subunit of ISWI complex, resulted in karyorrhexis and blocked cell cycle progression in AML cell lines [209]. BAF components of SWI/SNF complex have been confirmed to maintain an oligodendrocyte precursor cell (OPC)-like state in glioma stem cells, thereby providing novel candidates for targeted therapy in H3K27M-mutant gliomas [210]. CHD7 knockout inhibited tumor growth in an orthotopic mouse xenograft model of glioblastoma [211]. ARP5, component of INO80 complex, has also been described as a novel therapeutic target in glioblastoma due to its suggested oncogenic role in the disease [212]. Inhibition of BRG1 resolved differentiation blockade in fusion positive rhabdomyosarcoma cell lines [213], while depletion of CHD4, a coregulator of the oncogenic PAX3-FOXO1 transcription factor, resulted in reduced viability of fusion-positive but not of fusion-negative rhabdomyosarcoma in vitro [214]. ARID1B, a component of the SWI/SNF complex, has been described as a promising novel therapeutic target in ARID1A-mutant neuroblastomas [215].

## 7. Epigenetic Interventions in the Landscape of Anticancer Treatment 

Novel epigenetic drugs have been confirmed to influence the efficacy of other anticancer treatment modalities. Enhanced chemosensitivity, radiosensitivity and improved results of immunotherapy have been described in growing numbers of malignancies co-treated with epigenetic agents.

HMA treatment and depletion of SIRT6 resulted in increased sensitivity of AML cell lines to cytarabine treatment through the restored expression of BIK gene and inhibition of DNA repair process of double-strand breaks, respectively [216,217,218]. Loss of SIRT2 greatly enhanced chemosensitivity of AML cells harboring MLL-ENL fusion protein [219]. Inhibition of the demethylation of KMT enzyme G9a restored sensitivity of treatment-resistant B-ALL to glucocorticoid-induced cell death [220]. HDAC inhibitors may be applied to overcome rituximab resistance in B-cell lymphomas by the upregulation of CD20 expression on lymphoma cells [221]. Inhibition of HAT enzymes P300 and CBP sensitized mantle cell lymphoma to PI3K inhibitor idelalisib treatment in vitro and in vivo [222]. Knockdown of SIRT6 significantly potentiated the efficacy of doxorubicin in osteosarcoma cells [223]. Inhibition of EZH1/2 significantly increased the sensitivity of MYCN-amplified neuroblastoma cells to 5-fluorouracil therapy [224].

Adjuvant administration of decitabine resulted in a radiosynergistic effect in human medulloblastoma cell lines [225], while inhibition of PRMT6 enzyme improved the cytotoxic activity of radiotherapy against glioblastoma stem cells [226]. HMA therapy and entinostat have been described as promising radiosensitizing agents in embryonal rhabdomyosarcoma and PAX3-FOXO1 positive alveolar rhabdomyosarcoma cells, respectively [227,228].

Epidrugs can also prime antitumor immune response, which may give rise to the development of combination strategies with immunotherapy agents [229]. In AML cell lines and primary AML cells, BET inhibition improved antileukemia immunity by regulating PD-1/PD-L1 expression [230]. Decitabine plus anti-PD1 camrelizumab treatment increased the percentage of circulating peripheral central memory T-cells, which correlated with improved clinical response and survival outcome measures in R/R classical Hodgkin lymphoma [231]. Combined HDAC inhibitor and anti-PD-1 antibody treatment significantly promoted tumor regression and improved survival in a murine model of advanced soft tissue sarcoma [232]. Combination of anti-GD2 antibody and vorinostat was found to be highly effective in an aggressive orthotopic neuroblastoma model [233]. According to recently published in vitro results, combination of decitabine with CAR T-cell therapy is an attractive novel therapeutic approach to enhancing the tumor-specific killing of bladder cancer [234].

During the past few years, a growing amount of synergistic interactions of epigenetic agents have been identified. Combining LSD1 and JAK-STAT inhibition exerted synergistic antileukemic effects in Down syndrome-associated myeloid leukemia [235]. Combined treatment of promyelocytic leukemia cell line HL-60 with BRD inhibitor PLX51107 and vorinostat resulted in decreased cell proliferation and dramatically increased apoptotic rate [236]. EZH2 and HDAC inhibitors demonstrated potent synergy in lymphoma cell lines with EZH2 dysregulation [237]. BET-inhibitor JQ1 and CBP-inhibitor ICG-001 treatment synergistically inhibited proliferation and invasion potential in H3K27M-mutated DIPG cell lines [238]. CDK2 inhibitor milcilib synergized with BET inhibitor treatment in group 3 medulloblastoma in vivo and in vivo models [239]. In patient-derived xenograft models of embryonal rhabdomyosarcoma, synergistic growth inhibition was described in case of combinatorial treatment with entinostat and vincristine [240]. According to the results of a systematic review and meta-analysis published in 2018, a combination of HDAC inhibition and HMA therapy does not appear to be more effective and better tolerated than HMA alone in MDS and AML [241].

## 8. Conclusions and Future Perspectives

The major aim of the precision oncology treatment approach in the 21st century is to translate the revolution of molecular and genetic technologies into clinical practice. 

Reversible epigenetic alterations are novel hallmarks of cancer that often develop during early stages of malignant diseases. Due to their widespread interactions and involvement in the regulation of multiple biological processes, disruptions of epigenetic modifiers are considered as central hubs in the pathogenesis of cancer.

Patients should be profiled based on a systematic biological approach when diagnosed with a tumor. In addition to the identification of translocations and molecular genetic alterations, detailed evaluation of the epigenetic profile should be highlighted among future aims that could be completed with metabolomic characterization and the clarification of further, nonmolecular factors such as nutritional status and psychosocial condition.

Although precision medicine has also entered clinics for childhood tumors, major challenges should be noted. Epigenetic profiles of pediatric tumors are markedly different when compared to adult cancers. The biological impact of a certain epigenetic modifier enzyme depends highly on tissue microenvironment. The same chromatin regulator may harbor both tumor suppressant and oncogenic properties depending on the type of tumor. Genetic and epigenetic intratumor heterogeneity represents a remarkable challenge in treatment, contributing to tumor evolution, chemoresistance and relapse [20,237]. Special pharmacokinetic characteristics of different age groups within the pediatric population are to be considered.

Although epigenetic treatments are generally well-tolerated, toxicities and adverse events have also been registered, which are in need of further investigation. Prevention of late toxicities is of distinguished significance in pediatric cancer patients. Ensuring specificity is essential to avoid undesirable side effects, such as the activation of endogenous retroviral elements and long terminal repeats [242]. Azacitidine has been associated with aggravation of disease-associated thrombocytopenia [243], exacerbations of pre-existing crystal-induced arthritis [244] and development of pericardial effusion [245]. Further adverse events during epigenetic treatments have also been described, such as hypertriglyceridemia in a phase 1 study (NCT01321346) with panobinostat in pediatric leukemia and lymphoma patients [246]. Potential development of resistance and impairment of host response against viral replication should be noted [247,248]. 

In summary, targeted epigenetic interventions open new horizons in the treatment of childhood malignancies. Detailed genomic and epigenomic evaluation is required for the administration of patient-tailored combinations of epidrugs and conventional anticancer treatment modalities in the early stage of the disease. However, major challenges have to be resolved, epigenetic agents can contribute to improved survival outcomes, more efficient and less toxic treatment regimens, and improved quality of life for pediatric cancer patients. 

## Figures and Tables

**Figure 1 biomolecules-13-00061-f001:**
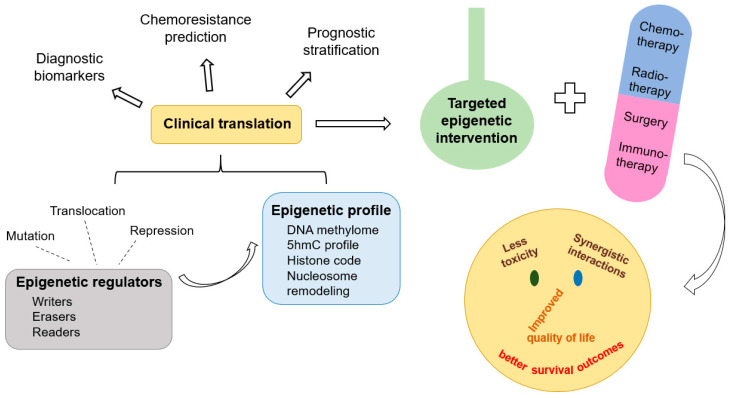
Epigenetic agents in the landscape of personalized anticancer treatment.

**Figure 2 biomolecules-13-00061-f002:**
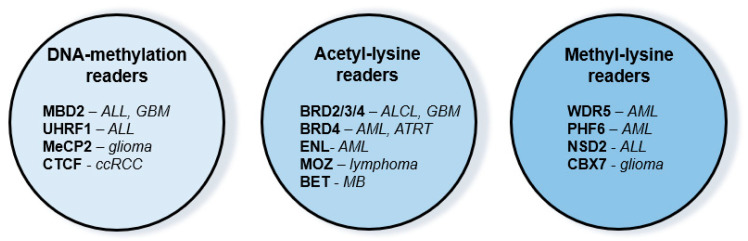
Reader molecules as novel therapeutic targets in pediatric malignancies (see references in the text). Abbreviations of diseases: ALCL: anaplastic large cell lymphoma, ALL: acute lymphoblastic leukemia, AML: acute myeloid leukemia, ATRT: atypical teratoid rhabdoid tumor, ccRCC: clear cell renal cell carcinoma, GBM: glioblastoma multiforme, MB: medulloblastoma.

**Figure 3 biomolecules-13-00061-f003:**
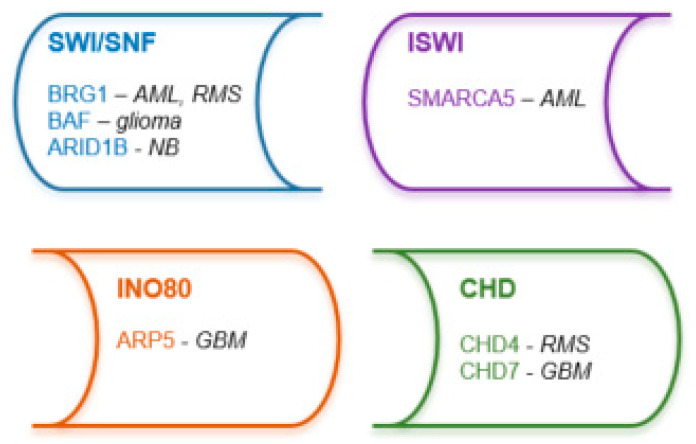
Targetable components of nucleosome remodeling complexes in pediatric malignancies (see references in the text). Abbreviations of diseases: AML: acute myeloid leukemia, GBM: glioblastoma multiforme, NB: neuroblastoma, RMS: rhabdomyosarcoma.

**Table 1 biomolecules-13-00061-t001:** Examples for epigenetic biomarkers in the differential diagnosis, prognosis and chemoresistance prediction of malignant diseases (see references in the text).

	Diagnosis	Prognosis	Chemoresistance Prediction
**DNA-methylation profile**	T-LBL, pilocytic spinal cord astrocytoma, retinoblastoma	JMML, adrenocortical tumors	Platinum chemotherapy (various cancers)
**TET2**		AML	
**5hmC pattern**	Metastatic neuroblastoma	AML, grade II astrocytoma	
**SIRT2**			Cytosine arabinoside and daunorubicin (AML)
**SIRT6**		Hodgkin lymphoma	
**EZH2**		MDS, ependymoma	Temozolomide (GBM)
**DOT1L**		ccRCC	
**SETD2**		MDS	
**NUP98/NSD1 fusion**		AML	

Abbreviations of diseases: AML: acute myeloid leukemia, ccRCC: clear cell renal cell carcinoma, GBM: glioblastoma multiforme, JMML: juvenile myelomonocytic leukemia, LBL: lymphoblastic lymphoma, MDS: myelodysplastic syndrome.

**Table 2 biomolecules-13-00061-t002:** Targeted epigenetic interventions in hematological malignancies and solid tumors (see references in the text).

Epigenetic Process	Enzyme Family	Enzyme Subfamily	Enzyme	Agent	Hematological Malignancies	Solid Tumors
**DNA methylation**	**DNMT**		DNMT1,3	Azacitidine, decitabine	MDS, AML, JMML, CMML, ALCL	GBM, neuroblastoma, RMS, synovial src, ccRCC, osteosarcoma, Ewing src, RT
**DNA hydroxymethylation**	**TET**		TET1,2	Bobcat339	T-ALL	Medulloblastoma, osteosarcoma
**Histone acetylation**	**HAT**			PU139 (pan-inhibitor)		Neuroblastoma
		** *p300/CBP* **	p300, CBP		MDS	
		** *MYST* **	MOF (MYST1)		NUP98-HOXA9-driven AML	
			HBO (MYST2)	WM-3835		Osteosarcoma
			TIP60			Osteosarcoma
		** *GNAT* **	GCN5		Pre-B ALL, non-APL AML, Burkitt lymphoma	
			PCAF			Alveolar RMS
**Histone deacetylation**	**HDAC**	**HDAC I, IIa, Iib, IV**	HDAC 1-11			
				Belinostat	APL	Retinoblastoma
				HDAC1i		Hepatoblastoma
				HDAC6i	Burkitt lymphoma	gr3 medulloblastoma, RMS, Ewing src
				I13	t(8;21) AML, MLL-rearranged AML	
				Mocetinostat	MLL-AF9	Glioma
				Panobinostat	R/R AML, relapsed cHL, t(4;11)-positive infant ALL	Soft tissue sarcoma, Wilms’ tumor
				Romidepsin	Burkitt lymphoma, t(4;11)-positive infant ALL	Neuroblastoma
				Spiruchostatin A		Neuroblastoma
				Trichostatin A	t(4;11)-positive infant ALL	Medulloblastoma
				Vorinostat/SAHA	R/R AML, t(4;11)-positive infant ALL	Medulloblastoma, glioma, grC ependymoma
**Histone deacylation**	**SIRT**		SIRT1	Tenovin 6, EX527	ALL, t(8;21) AML, MDS, Burkitt lymphoma	Glioma, soft tissue src, Ewing src
			SIRT2	Tenovin 6	ALL	Soft tissue src
			SIRT3			GBM
			SIRT4			Neuroblastoma, ccRCC
			SIRT5	NRD167	AML	
			SIRT6			Neuroblastoma, ccRCC, nasopharyngeal cc
			SIRT7			Glioma
**Histone Lys methylation**	**KMT**					
		**SET-domain**	EZH2	DZNep, tazemetostat (EPZ-6438), GSK343, EPZ005687	MDS, ALL, monocytic AML, NHL	SHH medulloblastoma, neuroblastoma, H3K27M gliomas, epithelial src, RMS, RT
			G9a (EHMT2)	BIX01294	ALL	Neuroblastoma, embryonal RMS, Ewing src
			NSD1		AML with t(5;11) translocation	
			SUV39H1			Astrocytoma, high-grade glioma, ccRCC
		**w/o SET domain**	DOT1L	Pinometostat (EPZ5676)	MLL-r AML, DNMT3A-mutant AML	Neuroblastoma, retinoblastoma
**Histone Arg methylation**	**PRMT**			AMI-1 and SAH (pan-inhibitors)		RMS
			PRMT1	28d	FLT3-ITD AML, MLL-r ALL	Medulloblastoma
			PRMT5		NHL	SHH medulloblastoma
			PRMT6	EPZ020411		GBM
**Histone demethylation**	**KDM1**		LSD1 (KDM1A)	FY56, S2157, HCI2509, SP2509	AML, T-ALL (CNS)	Ewing src, retinoblastoma
	**JHDM**		KDM3A (JHDM2A)			RMS, osteosarcoma
			KDM3C (JMJD1C)	JDI-10	MLL-r AML	
			KDM4A (JMJD2A)	SD49-7	LSC	
			KDM4B (JMJD2B)			Alveolar RMS
			KDM5A (JARID1A)		APL	
			KDM5B (JARID1B)	AS-8351		Ewing src
			KDM6A (UTX)	GSKJ4	AML	
			KDM6B (JMJD3)	GSKJ4	AML	
			JMJD6			ccRCC

Abbreviations of diseases: ALCL: anaplastic large cell lymphoma, ALL: acute lymphoblastic leukemia, AML: acute myeloid leukemia, APL: acute promyelocytic leukemia, ccRCC: clear cell renal cell carcinoma, CMML: chronic myelomonocytic leukemia, GBM: glioblastoma multiforme, JMML: juvenile myelomonocytic leukemia, LSC: leukemia stem cell, MDS: myelodysplastic syndrome, NHL: non-Hodgkin lymphoma, RMS: rhabdomyosarcoma, RT: rhabdoid tumor, src: sarcoma.

## Data Availability

No new data were created or analyzed in this study. Data sharing is not applicable to this article.

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
