# Peer review of "Targeted Epigenetic Interventions in Cancer with an Emphasis on Pediatric Malignancies"

_biomolecules, 2022, doi:10.3390/biom13010061_

Round 1
Reviewer 1 Report
1. Introduction: Do not start the sentence with a number
2. Figure 1: Do you mean Nucleosome conformation changes (remodeling)?
3. AML, OS, MDS, SAHA are to be expanded in the first place. Especially, OS for both overall survival and osteoblastic sarcoma has been used in the manuscript.
4. Correct as Wilm's tumors
5. Mention the Clinical trials identifier throughout the manuscript wherever studies on clinical trial studies are cited. It is difficult to identify the current status of the drug because lines such as "In a recently completed large phase II clinical study... (ref 87)" which was published 11 years ago, is conflicting and could not understand if there is any ongoing clinical trial.
6. Just like "HDACi" is mentioned, SIRT and EZH2 are to be denoted as SIRTi and EZH2i
7. The way pictures are presented is funny, but not attractive
8. Figures 2a and 2b are not interrelated.
9. While almost all classes of epigenetics-modifying drugs are discussed, the focus on pediatric malignancies keeps deviating except for rare citations of reports. It would be better to describe how these drugs could be useful for treating pediatric malignancies.
Author Response
- Introduction: Do not start the sentence with a number
Number „5” has been replaced with the word „five”.
Figure 1: Do you mean Nucleosome conformation changes (remodeling)?
The originally misspelled term has been modified to „nucleosome remodeling” in Figure 1.
AML, OS, MDS, SAHA are to be expanded in the first place. Especially, OS for both overall survival and osteoblastic sarcoma has been used in the manuscript.
AML, OS, MDS and SAHA are expanded in the revised version of the manuscript as follows:
“Mutations and translocations of epigenetic modifiers are frequently identified in cancer, such as TET2 and DNMT3A mutations, translocations of MLL and NUP98 genes in acute myeloid leukemia (AML), mutations of ARID1A and SMARCA4 genes in gliomas, SETD2 mutations in clear cell renal cell carcinoma (ccRCC), and oncohistone mutations in pediatric brain and bone tumours” (lines 40-44)
„Five year overall survival (OS) of children diagnosed with cancer has increased from 58% to 84.7% between the mid-1970s and mid-2010s, respectively” (lines 30-31)
„EZH2 mutation and SETD2 deficiency correlates with poor survival in myelodysplastic syndrome (MDS)” (lines 60-61)
“The first clinically successful HDAC inhibitor, suberoylanilide hydroxamic acid (SAHA/vorinostat) was approved by FDA in 2006 as a treatment for refractory or relapsed cutaneous T-cell lymphoma” (lines 184-186)
Abbreviation OS has been removed from Table 2. OS is used only for overall survival in current form of the manuscript.
Please note, that resulting from the insertion of an additional table (Table 1), numbering of tables has changed in the revised form of the manuscript.
Correct as Wilm's tumors
Wilms tumour has been replaced to Wilms’ tumour throughout the manuscript and in Table 2.
Mention the Clinical trials identifier throughout the manuscript wherever studies on clinical trial studies are cited. It is difficult to identify the current status of the drug because lines such as "In a recently completed large phase II clinical study... (ref 87)" which was published 11 years ago, is conflicting and could not understand if there is any ongoing clinical trial.
I absolutely agree with the reviewer. Since it was published 11 years ago, the words „recently completed” have been deleted from the above written sentence.
Clinical trial identifiers have been included in the revised text as follows:
“According to a phase 1 clinical trial (NCT01861002), published in 2018, azacitidine can be used safely in sequence with intensive chemotherapy in relapsed/refractory pediatric AML and offers encouraging clinical activity” (lines 104-106)
“According to a phase 1 clinical trial that was performed with panobinostat in 2020 (NCT02676323), 8/17 pediatric patients with relapsed or refractory (R/R) AML achieved complete remission (CR), and no dose-limiting toxicities were observed” (lines 189-191)
“In a phase I trial (NCT02412475), published in 2022, decitabine and vorinostat was well-tolerated and effective in R/R pediatric AML patients in combination with fludarabine, cytarabine, and G-CSF (FLAG) treatment” (lines 191-194)
„In a large phase II clinical study (NCT00742027) in patients with relapsed classical Hodgkin’s lymphoma, panobinostat reduced tumour measurements in 74% of patients, including 23% partial remissions and 4% CR” (lines 202-205)
“In a first-in-human, open-label, phase 1 study (NCT01897571), published in 2018, tazemetostat showed a favorable antitumour activity in refractory B-cell NHL and advanced solid tumours” (lines 287-289)
„Based on the positive results of a single-arm phase II basket study (NCT02601950), FDA granted accelerated approval to tazemetostat in 2020 for patients aged 16 years and older with metastatic or advanced epithelial sarcoma not eligible for complete resection” (lines 296-299)
“Based on the results of a phase 1 clinical trial in pediatric R/R leukemia patients with MLL-rearrangement, performed between 2014 and 2016 (NCT02141828), administration of EPZ5676 in combination with other antileukemia agents is warranted” (lines 322-324)
“Further adverse events during epigenetic treatments have also been described, such as hypertriglyceridaemia in a phase 1 study (NCT01321346) with panobinostat in pediatric leukemia and lymphoma patients” (lines 538-540)
Just like "HDACi" is mentioned, SIRT and EZH2 are to be denoted as SIRTi and EZH2i
Title of chapter 4.4.1 has been modified to EZH2i. Also, „EZH2 inhibitor” has been replaced to „EZH2i” in the following sentences:
„EZH2i 3-deazaneplanocin A induced growth inhibition and increased apoptotic rate in T-ALL Jurkat cells” (lines 280-281)
„EZH2i GSK343 led to significantly decreased viability, migration and invasion in neuroblastoma cell lines” (lines 294-295)
Since both inhibition, down-regulation, activation and overexpression of SIRT enzymes are included in the text, heading of chapter 4.3 has not been modified. However, „SIRT inhibitor” has been modified in the following sentence:
“SIRT1i molecules are promising therapeutic targets to treat metastatic disease in Ewing sarcoma by mediating tumour suppressive Notch response” (lines 259-261)
The way pictures are presented is funny, but not attractive + 8. Figures 2a and 2b are not interrelated.
Since they are not interrelated, Figure 2 and Figure 3 are included instead of Figure 2a and 2b in the revised version of the manuscript.
Both figures also have been modified in order to make them more attractive with a better quality of images too.
Please note that the numbering of figures have changed in the revised version of the manuscript.
While almost all classes of epigenetics-modifying drugs are discussed, the focus on pediatric malignancies keeps deviating except for rare citations of reports. It would be better to describe how these drugs could be useful for treating pediatric malignancies.
The focus on pediatric malignancies is explained in the introduction part of the revised manuscript as follows:
In this review, possibilities of targeted epigenetic interventions are discussed in the most common pediatric malignancies, including acute leukemias, lymphomas, neuroblastoma, gliomas, soft tissue sarcomas and kidney tumours. Vast majority of cited publications and clinical trials summarize experiences with pediatric and adolescent patients. However, since clinical experience can be first obtained in adult cancer patients (following successful in vitro and in vivo experiments), some data originating from cell lines, animal models and adult patients (NCT00742027 and NCT01897571) are also included in order to highlight the wide variety of novel opportunities. Major groups of epidrugs are discussed based on the epigenetic modification that they target.
Special considerations of applications of epidrugs in childhood (such as markedly different epigenetic profiles compared to adult cancers) are included in the discussion part, that have been completed with the following issues:
-„Special pharmacokinetic characteristics of different age groups within pediatric population are to be considered.” (lines 529-531)
-„Prevention of late toxicities is of distinguished significance in peditaric cancer patients.” (lines 533-534)
Reviewer 2 Report
This review concluded the advance in the both diagnosis and treatment of pediatric malignancies which targets epigenetic changes in clinics. Meanwhile, the author also elucidated the molecular mechanism of the relevant drugs. For the whole manuscript, it was well organized and presented the most of epigenetic alterations for some severe childhood cancers in detail. Of course, it will be important for applying some drugs aimed in individual treatment as well as for development of translation medicine.
There are some minor problems that still need author to resolve as following.
1. The author used many abbreviations showed the first time in Introduction part. Actually, this is very inconvenient for both readers and reviewers.
2. Introduction description (Line 55-87) seems quite redundant and complicated. It needs to simplified such as by using a table to list epigenetic marker changes for varied tumors.
3. The quality of Figure2b needs to be improved which seems quite obscure.
4. The author may better describe what the differences of the treatments between the childhood tumors and adult tumors in the aim of epigenetic target molecules.
Author Response
This review concluded the advance in the both diagnosis and treatment of pediatric malignancies which targets epigenetic changes in clinics. Meanwhile, the author also elucidated the molecular mechanism of the relevant drugs. For the whole manuscript, it was well organized and presented the most of epigenetic alterations for some severe childhood cancers in detail. Of course, it will be important for applying some drugs aimed in individual treatment as well as for development of translation medicine.
There are some minor problems that still need author to resolve as following.
- The author used many abbreviations showed the first time in Introduction part. Actually, this is very inconvenient for both readers and reviewers.
Abbreviations 2-HG, 5caC, ctDNA, DMP, DZnep, LBL, LSC, NGS, PADI4 and TCA have been removed.
- Introduction description (Line 55-87) seems quite redundant and complicated. It needs to simplified such as by using a table to list epigenetic marker changes for varied tumors.
Introduction part has been shortened and simplified. A novel table (Table 1) has been inserted that shows examples for epigenetic biomarkers with clinical translational significance.
Please note, that resulting from the insertion of an additional table (Table 1), numbering of tables has changed in the revised form of the manuscript.
- The quality of Figure2b needs to be improved which seems quite obscure.
Since they are not interrelated, Figure 2 and Figure 3 are included instead of Figure 2a and 2b in the revised version of the manuscript.
Both figures also have been modified in order to make them more attractive with a better quality of images too.
Please note that the numbering of figures have changed in the revised version of the manuscript.
- The author may better describe what the differences of the treatments between the childhood tumors and adult tumors in the aim of epigenetic target molecules.
Special considerations of applications of epidrugs in childhood (such as markedly different epigenetic profiles compared to adult cancers) are included in the discussion part, that have been completed with the following issues:
-„Special pharmacokinetic characteristics of different age groups within pediatric population are to be considered.” (lines 529-531)
-„Prevention of late toxicities is of distinguished significance in peditaric cancer patients.” (lines 533-534)
The focus on pediatric malignancies is explained in the introduction part of the revised manuscript as follows:
In this review, possibilities of targeted epigenetic interventions are discussed in the most common pediatric malignancies, including acute leukemias, lymphomas, neuroblastoma, gliomas, soft tissue sarcomas and kidney tumours. Vast majority of cited publications and clinical trials summarize experiences with pediatric and adolescent patients. However, since clinical experience can be first obtained in adult cancer patients (following successful in vitro and in vivo experiments), some data originating from cell lines, animal models and adult patients (NCT00742027 and NCT01897571) are also included in order to highlight the wide variety of novel opportunities. Major groups of epidrugs are discussed based on the epigenetic modification that they target.
Reviewer 3 Report
This is a very well written review which has been organized and described very succinctly. However, I do not think that the article is novel and reiterates several facts already mentioned in many other reviews. Maybe the author can focus more on current clinical molecules that are in trials or being developed instead of describing histone modifying enzymes in great detail as this is already well described in the field. I really like the table with clinical molecules that have been developed and tried for pediatric malignancies. Also, the figures can be improved. Maybe try using a publication illustrator software (like BioRender).
Author Response
This is a very well written review which has been organized and described very succinctly. However, I do not think that the article is novel and reiterates several facts already mentioned in many other reviews. Maybe the author can focus more on current clinical molecules that are in trials or being developed instead of describing histone modifying enzymes in great detail as this is already well described in the field. I really like the table with clinical molecules that have been developed and tried for pediatric malignancies. Also, the figures can be improved. Maybe try using a publication illustrator software (like BioRender).
In the revised version of the manuscript, some details about the previously well described molecular background of epigenetic modifiers and epidrugs have been removed:
„It is noteworthy that a broad range of non-histone proteins, tumour suppressors and oncogenes are also acetylated by HATs [66].”
„HDACs 1, 2, 3, and 8 belong to class I HDACs. HDACs 4, 5, 7, and 9 are class IIa HDACs. HDACs 6 and 10 are included in class IIb, and HDAC11 is the only member of class IV [79].”
„Nowadays, HDAC inhibitors compose a large family of novel anti-cancer agents, divided into four categories: (i) hydroxamic acids such as SAHA and panobinostat; (ii) cyclic peptides, including depsipeptide; (iii) benzamides, such as chidamide; and (iv) short chain fatty acids, including valproic acid [78].”
Introduction part has been shortened and simplified. A novel table about epigenetic biomarkers has been inserted. Major focus of this review has been clarified as follows:
“In this review, possibilities of targeted epigenetic interventions are discussed in the most common pediatric malignancies, including acute leukemias, lymphomas, neuroblastoma, gliomas, soft tissue sarcomas and kidney tumours. Vast majority of cited publications and clinical trials summarize experiences with pediatric and adolescent patients. However, since clinical experience can be first obtained in adult cancer patients (following successful in vitro and in vivo experiments), some data originating from cell lines, animal models and adult patients are also included in order to highlight the wide variety of novel opportunities. Major groups of epidrugs are discussed based on the epigenetic modification that they target.”
In order to get better focus on recent clinical experiences, identifiers of trials have been included in the revised text.
Since they are not interrelated, Figure 2 and Figure 3 are included instead of Figure 2a and 2b in the revised version of the manuscript. Both figures also have been modified in order to make them more attractive with a better quality of images too.
I absolutely agree that no novel data about clinical experiences are included. However, the major take home message of the review is to highlight the significance of individualized treatment based on a system biology approach in order to improve survival rates and life quality of pediatric cancer patients with less toxic and more effective anticancer treatment strategies. Major challenges and special considerations of applications of epidrugs in childhood (such as markedly different epigenetic profiles compared to adult cancers) are included in the discussion part, that have been completed with the following issues:
-„Special pharmacokinetic characteristics of different age groups within pediatric population are to be considered.” (lines 529-531)
-„Prevention of late toxicities is of distinguished significance in peditaric cancer patients.” (lines 533-534)
Please note that the numbering of figures and tables have changed in the revised version of the manuscript.
Round 2
Reviewer 1 Report
The manuscript is acceptable in the current format